# Cytomegalovirus-Associated Autoantibody against TAF9 Protein in Patients with Systemic Lupus Erythematosus

**DOI:** 10.3390/jcm10163722

**Published:** 2021-08-21

**Authors:** Yen-Fu Chen, Ao-Ho Hsieh, Lian-Chin Wang, Kuang-Hui Yu, Chang-Fu Kuo

**Affiliations:** 1Division of Rheumatology, Allergy and Immunology, Chang Gung Memorial Hospital, Linkou Center, Taoyuan 333, Taiwan; patrichen0693@gmail.com (Y.-F.C.); mayitsu2006@gmail.com (A.-H.H.); lian.chin.wang@gmail.com (L.-C.W.); 2Center for Artificial Intelligence in Medicine, Chang Gung Memorial Hospital, Linkou Center, Taoyuan 333, Taiwan

**Keywords:** systemic lupus erythematosus, cytomegalovirus phosphoprotein 65, TATA-box-binding protein associated factor 9, anemia, proteinuria

## Abstract

**Background**: Evidence indicates a causal link between cytomegalovirus (CMV) infection and the triggering of systemic lupus erythematosus (SLE). Animal studies have revealed that CMV phosphoprotein 65 (pp65) induces autoantibodies against nuclear materials and causes the autoantibody attack of glomeruli. IgG eluted from the glomeruli of CMVpp65-peptide-immunized mice exhibited cross-reactivity against dsDNA and TATA-box-binding protein associated factor 9 (TAF9). Whether the elevation of anti-TAF9 IgG is associated with anti-CMV reactivity in human lupus remains unclear. **Methods**: The sera from patients with rheumatic diseases, including ankylosing spondylitis (AS), gout, rheumatoid arthritis (RA), systemic lupus erythematosus (SLE), and Sjögren syndrome (SS) were examined using ELISA for antibodies of CMV, CMVpp65, and TAF9. **Results**: In total, 83.8% of the rheumatic patients had acquired CMV infections. The SLE patients had a high prevalence of anti-CMV IgM. The highest seropositivity rates for anti-HCMVpp65 and anti-TAF9 IgG were observed in the SLE patients. Purified anti-CMVpp65 IgG from CMVpp65/TAF9 dual-positive SLE sera reacted to both TAF9 and dsDNA. An increased prevalence of proteinuria and low hemoglobin levels were found in CMV IgG- and CMVpp65 IgG-positive SLE patients. **Conclusions**: This observation suggests that immunity to CMVpp65 is associated with cross-reactivity with TAF9 and dsDNA and that it is involved in the development of clinical manifestations in SLE.

## 1. Introduction

Systemic lupus erythematosus (SLE) is a chronic inflammatory immune disorder characterized by widespread loss of host immune tolerance to self-nuclear antigens. Increasing evidence indicates that human cytomegalovirus (CMV) infection precedes the onset of this loss of tolerance [1]. Most people infected with CMV during childhood are asymptomatic or exhibit self-limiting manifestations. Lifelong persistence, intermittent reactivation, and the dampening of the host defenses by interference with the host’s immune system are essential characteristics of CMV infection [2]. Several autoimmune disorders have been linked to CMV infection, particularly SLE [3]. Active CMV infections that are more frequently observed in SLE patients with active disease are related to the clinical manifestations and other complications [4]. CMV infection increases Ro/La protein expression on keratinocytes and laboratory cell lines [5,6]. The seroprevalence of anti-CMV and anti-U1 small nuclear ribonucleoprotein (snRNP) IgG is greater in SLE patients than healthy individuals, suggesting that CMV has a potential role in SLE development [7].

CMV phosphoprotein 65 (pp65), the major component of the extracellular viral particle, is responsible for the modulation of the host’s immune system during CMV infection [8,9]. SLE patients have a higher seroprevalence of anti-CMVpp65 IgG than other rheumatic patients [10]. Glomerular cellularity, proteinuria, and anti-CMVpp65 IgG cross-reacting with dsDNA and nuclear proteins were found in mice immunized with a CMVpp65 protein fragment [11,12]. Recently, we observed that the immunoglobulin G eluted from the glomeruli of CMVpp65-peptide-immunized mice bound to dsDNA and the human TATA-box-binding protein associated factor 9 (TAF9) protein [13]. To further confirm this observation in human subjects and investigate the association between serum anti-CMV/CMVpp65 IgG and the anti-TAF9 IgG antibody and SLE disease, cross-sectional research was performed on patients with one of five common rheumatic diseases.

## 2. Materials and Methods

### 2.1. Study Subjects

We followed the method proposed in our previous study [13]. All the participants were enrolled from rheumatology clinics in the Linkou branch of Chang Gung Memorial Hospital. The definition of rheumatic diseases was based on the American College of Rheumatology classification criteria, and a rheumatologist confirmed all the diagnoses. The current study was approved by the Institutional Review Board of Chang Gung Memorial Hospital, and all the participants provided written informed consent before their inclusion as required by the Declaration of Helsinki (approval numbers #201600798B0 and 201600795B0). The follow-up sera were collected from rheumatic patients whose disease was well-controlled at 3 months after disease onset. The clinical manifestation data were monitored and analyzed at the same time.

### 2.2. Antibody Purification

Antibody purification from serum was conducted as previously described [11]. Briefly, cyanogen-bromide-activated Sepharose (CnBr, 0.25 g/mL; Sigma Aldrich, St. Louis, Missouri, USA; CAS Number: 68987-32-6) was activated using 15 gel volumes of 1 mM HCl at 4 °C for 1 h. After activation, the CnBr resin was washed using an ice-cold coupling buffer twice. A total of 2 mg of CMVpp65 protein (ProSpect, Rehovot, Israeli; Catalog code: CMV-215) was dissolved by gentle rotation in ice-cold coupling buffer (0.1 M NaHCO_3_, 0.5 M NaCl, pH 8.3) with activated CnBr resin at 4 °C overnight. The free active groups on the CnBr resin were deactivated using 0.1 M Tris-HCl (pH 8.0) at room temperature for 2 h. After deactivation, the CnBr resin was washed three times with alternating buffer (0.1 M NaAc, 0.5 M NaCl, pH 4.0, and 0.1 M Tris-HCl, 0.5 M NaCl, pH 8.0) and washed with 10 mL of PBS once. For purification, 2 mL of serum from each patient with SLE and an anti-CMVpp65 IgG-positive status, in 20 mL of ice-cold PBS, was added to CMVpp65-protein-conjugated CnBr resin and incubated at 4 °C overnight. The bound antibodies were eluted using 1 mL of 0.1 M glycine (pH 2.0), and the eluted fraction was immediately neutralized with 50 μL of neutralizing buffer (1 M Tris-HCl, 2 M NaCl, pH 8.8).

### 2.3. ELISA Assay

The measurement of CMV IgM and CMV IgG was conducted using the Cytomegalovirus IgG/IgM ELISA Kit (Abnova, Taipei City, Taiwan; Catalog code: KA-1452 and KA-0228). The detection of serum CMV IgM and IgG was performed according to the manufacturer’s instructions. For the semi-quantitative determination of the CMVpp65 IgG, TAF9, and dsDNA IgG antibody titers, ELISA was conducted as previously described with minor modifications [13]. Briefly, 100 ng/well of CMVpp65 (Prospect, Rehovot, Israeli; Catalog code: CMV-215), TAF protein (MyBiosource, San Diego, CA, USA; Catalog code: MBS1385026), or purified calf thymus dsDNA (Sigma-Aldrich, St. Louis, MO, USA; CAS Number: 73049-39-5), in phosphate buffered saline (PBS, pH 7.4), was coated on a 96-well microtiter plate (Greiner Bio-One, North Carolina, USA; Catalog Number: 655074) at 4 °C overnight. After blocking the plate with 5% skimmed milk, 100× diluted human serum or 0.5 μg of purified IgG antibody in PBS was added, and the plate was incubated at 4 °C for 2 h. Following incubation, the unbound antibody was washed away four times using TBS with 0.05% Tween-20, and the bound antibody was incubated with 5000× diluted horseradish peroxidase (HRP)-conjugated anti-human IgG (Jackson ImmunoResearch, Pennsylvania, PA, USA; Catalog code: 109-035-088) at 4 °C for 2 h. After washing, TMB (Sigma-Aldrich, St. Louis, MO, USA) was added as the substrate, and the HRP activity was measured using a microplate ELISA reader at 450 nm (EZ read 400). Anti-CMVpp65 and anti-TAF9 positivity were defined according to the mean + 3SEM for the sera from the SLE patients.

### 2.4. CMV IgG Avidity Testing

The CMVIgG avidity was determined using a modified protocol [14]. The CMV IgG avidity was measured using the same CMV IgG kit with the use of TBST containing 8 M urea as a washing solution. We performed two sets of ELISA testing for CMV IgG avidity: one set was washed with TBST containing 8 M urea, and the other was washed with TBST without 8 M urea. The CMV IgG activities were used to calculate the avidity index (AI). The AI calculation was as follows: percentage of AI = (O.D._450_ absorbance value of CMV IgG activity per well with 8 M urea wash/O.D._450_ absorbance value of CMV IgG activity per well without 8 M urea wash) ×100. The cut-off value of AI >60% represented high IgG avidity, which is considered to indicate a past or recurrent infection, while the cut-off value of AI <30%, the low avidity, demonstrates primary infection.

### 2.5. Statistical Analysis

Statistical analyses of the titers and multiple-comparison corrections were performed using the GraphPad Prism software 8.0. Student’s *t* and two-tailed Fisher’s tests were used for these comparisons, with graphs depicting the mean ± SEM. Pearson’s correlation coefficient test was used to examine these comparisons, with graphs depicting the R squared and *p* values. The independent *t*-test, Kruskal–Wallis test, and chi-squared test were used to examine the results of the serological analysis. A *p* value <0.05 was considered significant, and different levels of significance are reported (* *p*  ≤  0.05; ** *p*  ≤  0.01; *** *p*  ≤  0.001).

## 3. Results

First, we examined the sera from patients with one of five rheumatic diseases—SLE (*n* = 193), SS (*n* = 70), RA (*n* = 84), gout (*n* = 92), and AS (*n* = 68)—for IgG and IgM antibodies against CMV (Table 1).

Over 75% of the patients of each rheumatic cohort who had acquired immunity from past CMV infections displayed anti-CMV IgG responses (Figure 1A,B) and exhibited intermediate or high CMV IgG avidity (Appendix A). SLE patients (10.9%, 21/193) had a higher rate of CMV IgM positivity than patients with other rheumatic diseases (Table 1).

In addition, the IgG antibody responses to CMVpp65 and the human TAF9 protein were evaluated using the sera from rheumatic patients with CMV IgG-positive statuses, including SLE (*n* = 163), SS (*n* = 64), RA (*n* = 76), gout (*n* = 70), and AS (*n* = 52). We found that the serum titers of anti-CMV IgG, anti-CMVpp65 IgG, and anti-TAF9 IgG were significantly more elevated in SLE patients than in the other rheumatic patients (Figure 1B and Figure 2A,B).

SLE sera (37.4% and 30.7%) exhibited the highest seropositive rate for anti-CMVpp65 IgG and anti-TAF9 IgG when compared to the sera from AS (32.7% and 11.5%) and SS (18.8% and 20.3%; Table 1). Moreover, a higher prevalence of coexisting anti-CMVpp65 and anti-TAF9 IgG was found in SLE (17.2%) than in AS (9.6%) and SS (12.5%; Table 1). Therefore, we purified anti-CMVpp65 antibodies from CMVpp65/TAF9 dual-positive SLE sera and examined antibody binding to dsDNA and the TAF9 protein (Figure 3).

ELISA results showed that the purified anti-CMVpp65 IgG could recognize dsDNA and the TAF9 protein. The positive association between anti-CMVpp65, anti-TAF9, and anti-dsDNA IgG antibody activities was significant in our cross-reactivity testing.

In addition, we compared the hematological and serological changes in SLE patients in the presence or absence of the IgG antibody to CMV and/or CMVpp65. CMV IgG-seropositive SLE patients had a higher prevalence of anti-dsDNA (*p* = 0.014) and anti-SSA (*p* = 0.014) antibodies than did CMV IgG-negative SLE patients. A statistically significant decrease in hemoglobin (*p* = 0.013) and hematocrit (*p* < 0.001) levels and an increased proteinuria rate (*p* = 0.026) were found in CMV IgG-seropositive SLE patients (Table 2). A high prevalence of proteinuria (*p* = 0.001) and low hemoglobin level (*p* < 0.001) were also observed in CMVpp65 IgG-positive SLE patients.

## 4. Discussion

In the current study, the CMV IgG seropositivity in 507 patients with rheumatic diseases was 83.8%. The CMV IgG seropositivity rate in rheumatic patients ranged from 76.1% to 91.4%, and few patients were CMV IgM positive. Our study population’s seropositivity rates for CMV IgG and IgM were close to those of previous studies [15,16]. Notwithstanding the observation that those patients with SLE or SS exhibited high seropositivity for CMV IgM, the CMV IgG antibody titer had already been elevated, as the patients were diagnosed with SLE or SS. The presence of anti-CMV IgM and IgG was assumed to indicate CMV reactivation rather than primary infection.

Antibody reactivity to the CMVpp65 antigen was less frequent in healthy individuals but more frequent in patients with SLE [10,12]. We observed a similar pattern with the antibody reactivity to the TAF9 antigen. The SLE patients had greater seropositivity for the anti-TAF9 IgG antibody than the patients with AS, gout, or RA. Compared with the groups of rheumatic patients, a higher prevalence of coexisting anti-CMVpp65 IgG and anti-TAF9 IgG was found in patients with SLE. Moreover, anti-CMVpp65 IgG purified from CMVpp65/TAF9 dual-positive SLE sera recognized CMVpp65, TAF9, or dsDNA, indicating that immunity to CMVpp65 is a potential trigger inducing cross-reactive antibodies in susceptible individuals.

During CMV infection or reactivation, the replication of CMV in kidney mesangial cells is implicated in the pathogenesis of CMV-induced renal disease [17,18]. The glomerular deposition of the CMV antibody–antigen complex is a critical factor contributing to immune complex glomerulonephritis [10,19]. Our previous study observed that BALB/c mice receiving the CMVpp65 peptide developed cross-reactive antibodies, proteinuria, and immunoglobulin deposition on the glomeruli [13]. In the present study, we found that the increased incidence of proteinuria in anti-CMVpp65-positive SLE patients was higher than that in anti-CMVpp65-negative SLE patients, suggesting that the elevated anti-CMVpp65 antibodies in SLE patients may be associated with renal damage and proteinuria. However, a causal link between autoimmune hemolytic anemia and SLE risk has also been discussed. CMV infection has been reported to inhibit erythropoietin production, which induces or exacerbates anemia in patients [20,21]. Low hemoglobin and hematocrit levels were found in SLE patients with CMV IgG or CMV/CMVpp65 IgG responses. However, the pathogenesis of anemia during CMV infection remains unclear, and our results cannot explain the causative link between the presence of CMV/CMVpp65 IgG and the occurrence of anemia in SLE. A comprehensive investigation is required to verify the role of anti-CMVpp65 IgG in anemia.

ELISA is a useful tool for determining the presence of serum anti-CMV IgM and IgG antibodies for the preliminary detection of CMV; however, several limitations of qualitative detection should be mentioned. For example, for patients with positive anti-CMV IgM results, it is not possible to clearly distinguish between CMV primary infection and reactivation. Furthermore, the time for the seroconversion of CMV IgM to CMV IgG or an elevated antibody response to CMVpp65 or the TAF9 protein varied between patients, meaning that quantitative testing is not suitable for monitoring the resolution of infection, especially when considering the fact that the seroprevalence rate for CMV is around 90% in Taiwanese patients [15]. Therefore, longitudinal research may offer comprehensive insights into the resolution of viral infection and cross-reactivity occurrence in autoimmune disease. Moreover, antibody reactivity toward linear epitopes is unable to be detected by ELISA testing. This may explain the low seropositivity of IgG antibodies against CMVpp65 in the current study [12].

## 5. Conclusions

In the present study, we examined the seroprevalence of anti-CMV IgG/IgM, anti-CMVpp65 IgG, and anti-TAF9 IgG among patients with five common rheumatic diseases. Compared to those from individuals with other rheumatic diseases, sera from SLE patients showed the highest seropositivity for antibodies against CMVpp65 and/or TAF9. The high proportion of coexisting serum IgG antibodies to CMVpp65 and TAF9 and the occurrence of cross-reactivity in SLE sera suggested that immunity to CMVpp65 is a potential trigger inducing cross-reactive antibodies. In addition, the high prevalence of proteinuria and low hemoglobin levels present in CMV IgG- and CMVpp65 IgG-positive SLE patients implied that CMV infection or reactivation might be involved in proteinuria and anemia during the development of SLE.

## Figures and Tables

**Figure 1 jcm-10-03722-f001:**
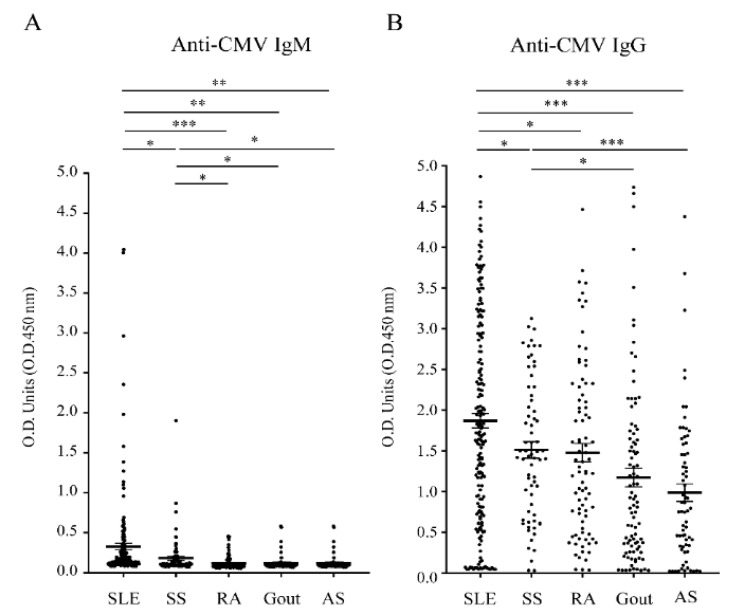
ELISA analysis of the antibody response to CMV using sera of patients with SLE (*n* = 193), SS (*n* = 70), RA (*n* = 84), gout (*n* = 92), and AS (*n* = 68). (**A**) Anti-CMV IgM activity. (**B**) Anti-CMV IgG activity. ***** *p*  ≤  0.05; ****** *p*  ≤  0.01; ******* *p*  ≤  0.001.

**Figure 2 jcm-10-03722-f002:**
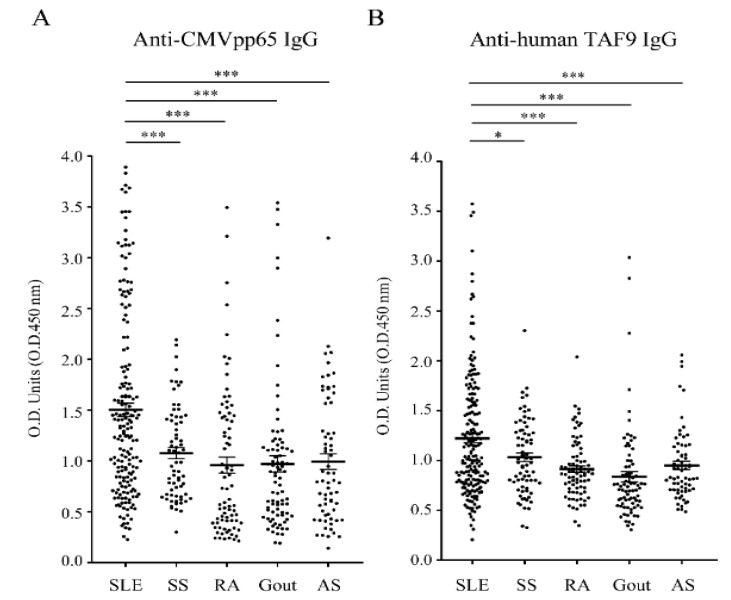
ELISA analysis of the antibody responses to CMVpp65 and TAF9 using sera of patients with SLE (*n* = 193), SS (*n* = 70), RA (*n* = 84), gout (*n* = 92), and AS (*n* = 68). (**A**) Anti-CMVpp65 IgG activity. (**B**) Anti-human TAF9 IgG activity. ***** *p*  ≤  0.05; ******* *p*  ≤  0.001.

**Figure 3 jcm-10-03722-f003:**
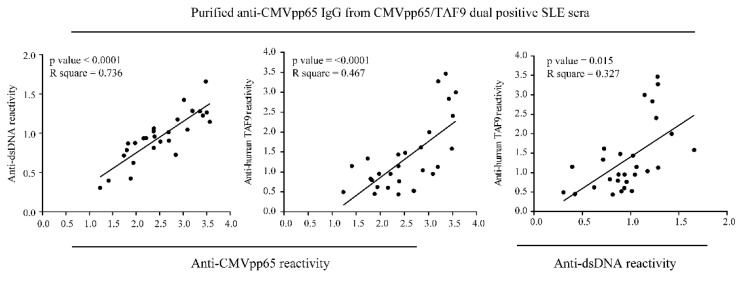
Analysis of the correlation between anti-CMVpp65 IgG, anti-human TAF9 IgG, and anti-dsDNA IgG by ELISA testing with anti-CMVpp65 IgG purified from CMVpp65/TAF9 dual-positive SLE sera (*n* = 28).

**Table 1 jcm-10-03722-t001:** Antibody responses to CMV, CMVpp65, and human TAF in rheumatic patients.

Characteristics	SLE (*n* = 193)	SS (*n* = 70)	RA (*n* = 84)	Gout (*n* = 92)	AS (*n* = 68)	*p* Value
Mean age, y (SD)	41.5 (12.7)	52.8 (11.1)	54.1 (12.6)	51.7 (14.1)	44.0 (12.8)	<0.0001 ^a^
Gender						<0.0001 ^b^
Male, *n* (%)	17 (8.8)	8 (11.4)	9 (10.7)	86 (93.5)	32 (47.1)	
Female, *n* (%)	176 (91.2)	62 (88.6)	75 (89.3)	6 (6.5)	36 (52.9)	
Anti-CMV IgG (+)	163 (84.5)	64 (91.4)	76 (90.5)	70 (76.1)	52 (76.5)	0.0063 ^b^
Anti-CMVpp65 IgG ^c^, *n* (%)	61 (37.4)	12 (18.8)	10 (13.2)	13 (18.6)	17 (32.7)	0.0004 ^b^
Anti-TAF9 IgG ^c^, *n* (%)	50 (30.7)	13 (20.3)	6 (7.9)	8 (11.4)	6 (11.5)	0.0001 ^b^
Dual positives ^c^, *n* (%)	28 (17.2)	8 (12.5)	3 (3.9)	4 (5.7)	5 (9.6)	0.0219 ^b^
Anti-CMV IgM	21 (10.9)	5 (7.1)	2 (2.4)	3 (3.3)	2 (2.9)	0.0217 ^b^
Anti-CMV IgM and IgG	21 (10.9)	5 (7.1)	2 (2.4)	3 (3.3)	2 (2.9)	0.0217 ^b^

SLE, systemic lupus erythematosus; SS, Sjögren syndrome; RA, rheumatoid arthritis; AS, ankylosing spondylitis; ^a^ Kruskal–Wallis test. ^b^ Chi-squared test. ^c^ The positive sera from patients with anti-CMV IgG antibodies. The anti-CMVpp65 and anti-TAF9 positive sera were defined according to the mean + 3SEM of the sera from SLE patients. SD, standard deviation; y, year.

**Table 2 jcm-10-03722-t002:** Comparison of hematological and serological abnormalities in SLE patients in the presence or absence of IgG antibody to CMV and CMVpp65.

Characteristics	CMV IgG Negative (*n* = 30)	CMV IgG Positive (*n* = 163)	*p* Value	CMVpp65 IgG Negative (*n* = 102)	CMVpp65 IgG Positive (*n* = 61)	*p* Value
Mean age, y (SD)	43.4 (12.7)	42.9 (12.1)	0.837 ^a^	42.9 (12.3)	42.8 (12.9)	0.961 ^a^
Female, *n* (%)	26 (86.7)	150 (92.0)	0.309 ^b^	94 (92.2)	56 (91.8)	1.000 ^b^
Cre, *n* (%)	3 (10.0)	25 (15.3)	0.580 ^b^	16 (15.7)	9 (14.8)	0.873 ^c^
Urine protein, *n* (%)	3 (10.0)	48 (29.4)	0.026 ^c^	21 (20.6)	27 (44.3)	0.002 ^c^
WBC, *n* (%)	7 (23.3)	24 (14.7)	0.278 ^b^	14 (13.7)	10 (16.4)	0.642 ^c^
RBC, *n* (%)	8 (26.7)	52 (31.9)	0.569 ^c^	29 (28.4)	23 (37.7)	0.219 ^c^
Hb, *n* (%)	9 (30.0)	89 (54.6)	0.013 ^c^	42 (41.2)	47 (77.1)	<0.0001 ^c^
Hct, *n* (%)	10 (33.3)	127 (77.9)	<0.001 ^c^	78 (76.5)	49 (80.3)	0.566 ^c^
MCV, *n* (%)	4 (13.3)	29 (17.8)	0.551 ^c^	18 (17.6)	11 (18.0)	0.950 ^c^
MCH, *n* (%)	3 (10.0)	34 (20.9)	0.165 ^c^	19 (18.6)	15 (24.6)	0.365 ^c^
MCHC, *n* (%)	2 (6.7)	14 (8.6)	1.000 ^b^	5 (4.9)	9 (14.8)	0.029 ^c^
Platelets, *n* (%)	2 (6.7)	26 (16)	0.262 ^b^	13 (12.7)	13 (21.3)	0.148 ^c^
Low C3, *n* (%)	11 (36.7)	73 (44.8)	0.410 ^c^	40 (39.2)	33 (54.1)	0.064 ^c^
Low C4, *n* (%)	8 (26.7)	69 (42.3)	0.107 ^c^	42 (41.2)	27 (44.3)	0.699 ^c^
Autoantibodies						
Anti-dsDNA, *n* (%)	12 (40.0)	104 (63.8)	0.014 ^b^	63 (61.8)	41 (67.2)	0.484 ^c^
Anti-SSA, *n* (%)	4 (13.3)	59 (36.2)	0.014 ^c^	36 (35.3)	23 (37.7)	0.757 ^c^
Anti-SSB, *n* (%)	6 (20.0)	17 (10.4)	0.214 ^b^	11 (10.8)	6 (9.8)	0.848 ^c^
Anti-Sm, *n* (%)	6 (20.0)	31 (19)	0.900 ^c^	19 (18.6)	12 (19.7)	0.869 ^c^
Anti-RNP, *n* (%)	3 (10.0)	34 (20.9)	0.165 ^c^	20 (19.6)	14 (23.0)	0.611 ^c^
Anti-Scl70, *n* (%)	0 (0)	1 (0.6)	1.000 ^b^	0 (0)	1 (1.6)	0.374 ^b^
Anti-Jo1, *n* (%)	0 (0)	2 (1.2)	1.000 ^b^	1 (1)	1 (1.6)	1.000 ^b^
Anti-CentB, *n* (%)	1 (3.3)	4 (2.5)	0.575 ^b^	3 (2.9)	1 (1.6)	1.000 ^b^
Anti-Histone, *n* (%)	4 (13.3)	22 (13.5)	1.000 ^b^	14 (13.7)	8 (13.1)	0.912 ^c^

Normal adult reference ranges. WBC (leukocyte count, M: 3.9~10.6, F: 3.5~11, ×10^9^/L), RBC (erythrocyte count, M: 4.5 ~ 5.9, F: 4.0~5.2, ×10^12^/L), Hb (hemoglobin, M: 13.5~17.5, F: 12~16, g/dL), Hct (hematocrit, M: 41~53, F: 36~46%), MCV (mean corpuscular volume, 80–100, fL), MCH (mean corpuscular hemoglobin, 26–34, pg), MCHC (mean corpuscular hemoglobin concentration, 31–37%), platelet count (150–400, ×10^9^/L), Cre (creatinine, M: 0.64~1.27, F: 0.44~1.03, mg/dL), urine protein (>30 mg/dL), C3 (complement C3, 80–155, mg/dL), C4 (complement C4, 13–37, mg/dL). M: male; F: female. Anti-SSA, anti-Sjögren syndrome-related antigen A antibody; Anti-SSB, anti-Sjögren syndrome-related antigen B antibody; Anti-Sm, anti-Smith antibody; Anti-RNP, anti-nuclear ribonucleoprotein antibody; Anti-Scl70, anti-topoisomerase I antibody; Anti-CentB, anti-centromere B antibody. ^a^ Independent *t* test; ^b^ Fisher’s exact test; ^c^ Chi-square test.

## Data Availability

The datasets analyzed during the current study are not publicly available due to our IRB policy, but they are available from the corresponding author upon reasonable request.

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
