# Peer review of "Cytomegalovirus-Associated Autoantibody against TAF9 Protein in Patients with Systemic Lupus Erythematosus"

_jcm, 2021, doi:10.3390/jcm10163722_

Round 1

Reviewer 1 Report

The authors give a follow-up of previous observations about the presence of anti-CMV cross-reacting autoantibodies in different rheumatic diseases with emphasis on SLE.

Methodology : the authors compare the results of purified anti-CMV65pp IgG with dsDNA and anti-TAF9 autoantibodies. They however do not mention how they purified the IgG. This must be added in the methodology section.

In Table 2, the 193 SLE patients are reduced to 163 in the anti-CMV65pp results. What happened to the 30 patients who disappeared?

Overall, the english is at the border of uncomprehensibility. The text should be rewritten by an english ediitor.

Author Response

Dear reviewer 

Sincerely

Ao-Ho Hsieh

Reviewer 2 Report

While the manuscript is well written, I would like to clarify some points before we can accept their conclusions .  

Methods: Concerning the timeframe of the antibody determination, when were these antibodies tested? At the onset? At the same time? During the follow-up? (if so, when?)

The authors tested CMV IgG avidity but data are not shown in the Results section. 
Discussion:   One limitation of the study is that CMV reactivation can only be established by viral load (PCR) testing, as the authors already acknowledge in the manuscript. On the other hand. ELISA is a helpful tool to detect the presence of these antibodies but their pathogenic role remains uncertain (i.e. as TAF9 is a transcription factor EMSA testing may elucidate if DNA binding is impaired or not)a and CMV, as CMV may cause anemia throughout myelopoiesis interference (a central mechanism) while haemolytic anemia is driven by peripheral mechanism. In addition TAF9 is required for optimal B-globin expression Sengupta et al (PNAS 2009)  (DOI: 10.1073/pnas.0808347106, which reflects an alternative cause for anemia in the context of autoantibody production, The rationale to establish the link with this study is difficult to justify and I would consider rephrasing or even deleting the statement.

Author Response

Dear reviewer

Sincerely

Ao-Ho Hsieh

Round 2

Reviewer 1 Report

The authors correctly answered my questions and substantially improved the english text.